# Robustness Optimization of an Existing Tablet Coating Process Applying Retrospective Knowledge (rQbD) and Validation

**DOI:** 10.3390/pharmaceutics12080743

**Published:** 2020-08-07

**Authors:** Albert Galí, Magda Ascaso, Anna Nardi-Ricart, Marc Suñé-Pou, Pilar Pérez-Lozano, Josep M. Suñé-Negre, Encarna García-Montoya

**Affiliations:** 1PhD collaborator at the Pharmacy and Pharmaceutical Technology and Physical Chemistry Department, University of Barcelona, 08028 Barcelona, Spain; agaliser@hotmail.com (A.G.); magda.ascaso@hotmail.com (M.A.); 2Department of Pharmacy and Pharmaceutical Technology and Physical Chemistry, Faculty of Pharmacy and Food Sciences, University of Barcelona, Av. Joan XXIII, 27–31, 08028 Barcelona, Spain; msunepou@gmail.com (M.S.-P.); perezlo@ub.edu (P.P.-L.); jmsune@ub.edu (J.M.S.-N.); encarnagarcia@ub.edu (E.G.-M.); 3Pharmacotherapy, Pharmacogenetics and Pharmaceutical Technology Research Group Bellvitge Biomedical Research Institute (IDIBELL), Av. Granvia de l’Hospitalet, 199–203, 08090 L’Hospitalet de Llobregat, Spain

**Keywords:** retrospective data, design of experiments, critical process parameters, coating optimization, coating defects, process validation

## Abstract

The objective of these studies is to verify and validate the improvement in the inter-tablet coating uniformity for an industrially commercialized coated tablet, without involving changes in the approved registration dossier. Using the CPP (critical process parameters) determined from previous retrospective statistical analysis, the recommended working ranges are identified. Retrospective analysis showed that the design of experiments (DoE) provided an improved process variable configuration. Therefore, it is decided to study two critical parameters: Product temperature and drum speed, with an additional 2^2^ experimental design. The quality results of the samples analyzed show that the aesthetic defects of the batches made with the new working ranges have been reduced. These results have also been corroborated with the 42 industrial batches manufactured with the new ranges. With the optimized parameters, tablets have been coated and the suitability of the model determined. The results demonstrated the overall reliability and effectiveness of the proposed Quality by Design approach and provides a useful tool to help optimize the industrial coating process. This study confirms that it is possible to optimize and validate the manufacturing process of an existing commercial product by means of a DoE with retrospective data. Therefore, no variation in the dossier is required.

## 1. Introduction

One of the questions that is often asked by the pharmaceutical industry is how it can improve the quality of its commercial products.

GMP (Good Manufacturing Practices) and ICH Q10 talk about the continuous improvement and implementation that are expected throughout the product lifecycle to facilitate innovation and continual improvement in the pharmaceutical industry.

One of the reasons for not considering improving the industrial processes of commercialized products is the registration dossier approved by the authorities.

The optimization of tablet-coating processes generally warrants extensive experimental research. The hypothesis of this study is, however, to ascertain if improvements can be made without major investments in resources or additional tests at the galenic phase that would ultimately translate into more efficient commercial manufacturing processes. In other words, it is not necessary to go back to the galenic development phase in order to optimize commercial processes, as improvements could be proposed based on historical data from the industrial batches already manufactured. With a central composite design considering independent and dependent variables, a model could be obtained to achieve an optimized response characteristic for the three surface defects. Such surface defects can be classified into the following three groups: Erosion, white spots, and poor coating uniformity. The aim, in this case, is to apply the methodology to a complex process, i.e., tablet coating. As an example, Puñal Peces et al. already established a design space for a wet granulation and a compression process retrospectively [1]. A design of experiments (DoE) with historical data from four years of industrial production was carried out using, as experimental factors, the results of the previous risk analysis and eight key parameters (quality specifications) that encompassed process and quality control data. Experience showed that it is possible to determine design spaces retrospectively [2], with the greatest difficulty being the handling and processing of high amounts of data. However, the practicality of this study is very interesting as it facilitates a design space with minimal investment in experiments as the actual production batch data are processed statistically.

Although tablet-coating processes are widely used in the pharmaceutical industry, they often lack adequate robustness. In a coating process in particular, the coating uniformity is a critical quality attribute. To comply with the acceptance value of the coating layer, process understanding is essential and experimental investigations are crucial to obtain a good understanding of it. In the context of quality by design (QbD), coating process simulations can be used as a tool to enhance process understanding.

In order to perform these simulations, the impact of the process variables, the process formulation parameters, and the quality of the product must be evaluated.

However, the evaluation of critical process variables that impact the coating uniformity and the validation of the model predictions require experimental studies.

A coating process consists of simultaneous spraying, mixing, and drying processes. These components and their respective parameters have to be considered with regard to the coating uniformity. In general, the coating variability can be influenced by the tablet movement in the coater and by the spray properties [3]. Several authors evaluated critical process parameters on the inter-tablet coating uniformity by statistical design of experiments (DoE). Therefore, establishing a design space [2] for the coating step could prove very helpful in ensuring reproducible quality results whereby tablet appearance, biopharmaceutical characteristics (such as disintegration and dissolution), and, consequently, therapeutic activity would be unaffected.

A product already available on the market as coated tablets (cytostatic agent, 25 mg of active ingredient per dose, weight of the tablet 100 mg) showing problems associated with its coating was selected for the quality improvement (from a technological point of view) study. This cytostatic drug was studied in a previous preliminary phase [4] as an agent with the highest number of quality-related problems, and the number of commercial batches was 36 to carry out a retrospective statistical study. Preliminary studies indicated that individual parameters affected the quality of the coated tablets.

The aim of this study is to detect critical process parameters retrospectively and to evaluate those that significantly impacted the studied responses (defects), which, in this case, are erosion and white spots on the coated tablet. The intention is to establish an appropriate operating range, as other authors did before in a prospective way for a coating process [5,6,7,8,9,10] or in a retrospective way for others pharmaceutical forms [11,12] for the majority of the parameters evaluated to ensure the process robustness.

From a technological viewpoint, the tablet-coating process has been optimized to reduce aesthetic-type quality defects without introducing any changes to the manufacturing process.

## 2. Materials and Methods

### 2.1. Materials

The materials used are those listed under the standard formula of the medicine marketed since 2007 but which are not specified here for reasons of confidentiality. The batch size is an industrial batch of 250,000 tablets manufactured according to a wet granulation process, followed by tablet compression and coating processes (see Figure 1).

### 2.2. Method to Determine the Range of the 2 Critical Parameters of Coating

With the results obtained in this preliminary phase (see Table 1), an experimental design is carried out with two of the critical parameters detected (drum speed and product temperature) to establish the optimal operating range when applied to a commercial batch size, from which the optimal process conditions can be defined following a statistical data analysis. Later, validation batches will then be used to confirm the data obtained.

These batches are formulated without the active ingredient (placebo batches).

It has been considered preferable to make batches with the same size as a commercial batch rather than a reduced batch with the active ingredient.

In this way, the batches are as representative as they would be on a commercial scale, manufactured with the same equipment and presenting the same galenic characteristics.

The percentage of the API (Active Pharmaceutical Ingredient) in the tablet is 25 mg in a total weight of 97 mg, meaning 25.7%.

Finally, in the last phase, the process conditions detected with the factorial design are confirmed with the manufacture of 3 industrial-size validation batches of the cytostatic drug. The purpose of this phase is to confirm that the manufacturing of the placebo batches and that the proposed operating ranges are suitable for the commercial batches of this product.

### 2.3. Method Used to Monitor Defects

To evaluate and quantify tablet defects, a visual inspection is carried out at the end of the coating process according to the ANSI/ASQ Z1.4-2003 standard (Sampling Procedures and Tables for inspection by Attributes) [13].

The sampling plan parameters are: Inspection type: Normal. Sample size: Level II, single sampling. Acceptance quality limit (AQL): Unacceptable defects 0.015. Critical defects 0.1; Major defects 1.5; Minor defects 2.5.

A sample of 800 coated tablets is taken from each validation batch with a visual inspection according to Level II of the ANSI/ASQ Z1.4-2003 standards.

The acceptance and rejection (AQL) criteria for the sample size can be consulted in the corresponding sampling plans for normal inspections according to the ANSI/ASQ Z1.4-2003 standard. For a batch size of 150,001–500,000 tablets, the P sampling letter applies, which corresponds to a sample size of 800 tablets. The acceptance quality levels (AQL) for normal inspections are:

AQL 0.015Accepted = 0Rejected = 1 (N = 800)AQL 0.1Accepted = 2Rejected = 3 (N = 800)AQL 1.5Accepted = 21Rejected = 22 (N = 800)AQL 2.5Accepted = 21Rejected = 22 (N = 500)

According to the ANSI/ASQ Z1.4-2003 sampling plan for normal inspections, a sample size of 800 tablets is needed for unacceptable defects (AQL 0.015) and of 500 tablets for minor defects (AQL 2.5). However, as the objective of the study is the validation of a non-functional coating, a sample size of 800 tablets is considered adequate for all defects, this being considered the strictest case, so, if it is met, it will mean that the batch is compliant in terms of acceptable defects.

## 3. Results and Discussions

The results obtained in the preliminary retrospective analysis allows an operating range to be established for most of the parameters studied, see Table 1 [4]. As other authors have proposed previously [6,7,8,9,10,14,15], it would possible to have a mathematical model to predict critical process parameters and to monitor the coating process. However, in the present work, the main difference is that the prediction has been made by a retrospective factorial design with data of 36 batches.

Having detected the critical process parameters in the retrospective statistical study, an optimal value can be set for most of these parameters, without the need to study other variables. For the nine factors that could affect the tablet-coating process (see Table 1), minimum operating values are defined for all nine, maximum values are defined for eight, and optimum or objective values for five. For two of the undefined values (product temperature and drum speed), it is considered that it could be optimized by conducting a prospective detailed analysis of the experimental study according to a new 2 × 2 factorial design. The speed of drum 1 is selected for the study, considered the most critical because, at this stage, there is a much higher risk of tablet erosion and it is a process parameter that is directly related to erosion, as observed by Just, S. et al. [14].

In this DoE, four commercial-scale batches (tablets) are manufactured with all possible combinations of both parameters. A fifth batch is also manufactured with central conditions, i.e., setting the two parameters at an intermediate value relative to the other batches. The factorial design tests are described in Table 2, including the parameter combinations per batch. The five commercial-scale batches have been manufactured accordingly.

Table 3 compares the mean values obtained in each batch of the factorial design that are within the previously established specifications. As can be seen, the mean product temperatures in each batch remained within the range established in the factorial design, with the inlet and product temperatures remaining stable throughout the process.

The main difference between the batches is the length of the process, with this being longer in the last three batches manufactured (AGS.141217.01, AGS.141218.01. and AGS.141218.02). These differences are mainly due to interruptions in the process following the misreading of the equipment’s flow meter. Evidently, the higher the number of interruptions, the longer the process, as can be seen with batch AGS.141208.02, which was interrupted up to five times. However, as these are short (approximately 5 s) and controlled interruptions, no impact on the quality of the product is expected.

As already concluded in the previous retrospective study of the batches, the greater the suspension added, the fewer the tablet defects detected [4]. The aim with these batches is to apply the maximum theoretical quantity of suspension, avoiding values below this; in some cases, it is even slightly exceeded, but always remaining within the established limits (see batch 3 in Table 3).

A sample of 500 coated tablets was taken from each batch (five commercial-scale batches, details of process in Table 2) for visual inspection to determine the number of defective tablets per batch. The results are shown in Table 4. As can be observed, there were still some coated tablets with some degree of erosion and/or white spots, but the coating uniformity was optimized in all batches and no between-batch differences were observed.

To evaluate the defects (erosion and white spots), this factorial design was analyzed statistically using the Statgraphics 5.1 software. The Pareto charts corresponding to each study are shown in Figure 2 and Figure 3. The lack of significant impact by parameters on the defects (erosion and white spots) is clearly observed. The pink bars in Figure 2 and Figure 3 show the parameters that are directly proportional to the responses, i.e., increasing the value of the parameter also implies an increased effect on the response. Conversely, the red bars in Figure 2 and Figure 3 represent the parameters that are inversely proportional to the responses in which the effect on these responses decreases as the value of the parameter increases.

The factorial design studies the effects of two factors in five trials. It was executed in a single block and the order of the experiments was totally randomized, which protects against the effects of hidden variables. For this study, only 1 degree of freedom was available to estimate the experimental error, as no repetitions were possible, as these were industrial batches. It is generally recommended to include enough central points in a factorial design to allow, at least, 3 degrees of freedom for the error, which is not possible in this case, due to product limitations. However, one central point was considered sufficient for the study.

The surface plot shown in Figure 4 traces the ascending (or descending) influence of the factors on the defect being studied, generating a response surface in which the estimated response varies rapidly with minimum variation in the experimental factors. These would be good locations for additional experiments should we wish to increase or decrease erosion. Six points were generated by changing the product temperature in increments of 0.4. As can be clearly seen in Figure 4, by decreasing both the product temperature and drum speed, the number of coated tablets with erosion also decreases. Analysis of variance gives an R^2^ correlation of 97.3% (more data and graphs are accessible in the full paper [16]). The product temperature also affects tablet erosion in that the lower the temperature, the lower the erosion of the tablets. However, the impact is lower and slight variations do not have a significant effect.

Figure 5 shows the surface contour plot used to identify the most appropriate operating area to minimize the response studied (erosion). In this case, like in Figure 3 and Figure 4, the working area to reduce the number of eroded tablets is defined by a lower product temperature and a lower drum speed (shown in red in Figure 5). Taking into account the results of this statistical analysis of the data obtained in the factorial design, the best process conditions to reduce tablet erosion are those established for batch AGS.141201.01: Lower product temperature (44 °C) and drum speed (7 rpm in the initial phase followed by 10 rpm until the end of the process).

The surface plot shown in Figure 6 traces the ascending (or descending) influence of the factors on the defect being studied, generating a response surface in which the estimated response varies rapidly with minimum variation in the experimental factors. These would be good locations for additional experiments should we wish to increase or decrease the number of white spots. Six points were generated by changing the product temperature in increments of 0.4. Analysis of variance gives an R^2^ correlation of 94.0%, more data and graphs are accessible in the full paper [16]. As can be clearly seen in Figure 6, by decreasing both the product temperature and drum speed, the number of coated tablets with white spots also decreases. The product temperature also affects the number of tablets with white spots in that the lower the temperature, the lower the effects of this defect. However, the impact is lower and slight variations do not have a significant effect.

Figure 7 shows the most appropriate working area to minimize the response studied (white spots). As was the case with erosion (see Figure 5), the working area that would produce coated tablets with fewer white spots is that with the lower product temperature and lower drum speed (as shown in red in Figure 7). Given the results of this statistical analysis of the data obtained in the factorial design, the best process conditions to reduce the number of white spots in the coated tablets are those established for batch AGS.141201.01: Lower product temperature (44 °C) and drum speed (7 rpm in the initial phase followed by 10 rpm until the end of the process). Therefore, both statistical studies confirm that the conditions established for batch AGS.141201.01 are optimal to reduce the number of defects (erosion and white spots) in the coated tablets. Therefore, these conditions will be established for the commercial validation batches.

### 3.1. Validation of the Coating Process

From the optimal conditions established after the statistical treatment of the results of the factorial design, the coating process (in O’Hara Labcoat IIX coating system) had to be validated. Three validation batches were therefore manufactured, on a commercial scale (tablets), with the optimal parameters according to the preliminary multivariate study, a retrospective statistical analysis, and the later 2^2^ prospective factorial design (see Table 5). These batches would confirm that the established process parameters were optimal to improve the appearance of the coated tablets.

During the coating process, in-process controls of the weight gain (50 tablets per test) were carried out periodically (with each 500 g of suspension added).

Table 6 shows that the erosion and coating uniformity defects have been fully optimized, with no tablet with these defects detected. Nonetheless, some white spots still appeared in some of the tablets during the visual inspection, but the spots were smaller and more difficult to detect, and the quantity detected was within specifications.

### 3.2. Factorial Design

With the results obtained in the DoE (see Table 4), it was observed that the higher the drum speed, the greater the number of eroded tablets, although Just, S. et al. [14] found the opposite effect. However, the lower drum speed set in the factorial design was higher than the values obtained from the retrospective historical [15], which confirms the result of the retrospective study, concluding that the drum speed should be increased to improve coating uniformity and number of white spots [17].

A very significant improvement was observed in the inter-tablet and intra-tablet coating uniformity in all batches, as was previously observed by Tobiska, S. et al. [18], Tobiska, S. et al. [19]. In the samples selected for the visual inspection, tablets with this defect were not detected (see last columns in Table 4).

The parameters established for batch AGS.25.141201.01 produced the least number of tablets with aesthetic defects, with only 1 coated tablet found with white spots out of the sample of 500 tablets. Therefore, the combination of a low product temperature (around 44 °C) and a low drum speed (7 rpm up to 2000 g of sprayed suspension, then 10 rpm) produced coated tablets with fewer appearance defects. These are, therefore, the values set for these parameters for the coating process validation batches. The values are described in Table 5.

### 3.3. Validation of the Coating Process

Firstly, the minimum amount of suspension applied to the tablets in these batches was that established theoretically, and this produced better results, confirming that the coating process should be stopped once the full theoretical suspension quantity has been applied.

Secondly, longer coating processes have been shown to be more beneficial for the product as fewer defects were observed given the greater homogeneity, corroborating the findings of different authors [10,20]. Nonetheless, coating times are not easily defined and can only be controlled by the spray flow, which is reduced to 45 g/min in order to slightly lengthen the process.

Product temperature is one of the main parameters that was adjusted based on the results of the factorial design. It was kept at about 44 °C throughout the process, allowing the polymer film to form accordingly. The inlet temperature was also adjusted so that the desired product temperature could be reached.

The pre-heating time was set at 4 min to be as short as possible to avoid erosion of the tablet cores and to improve the uniformity of the coating color. To ensure that the cores reached the required temperature before starting the spraying process, the vacuum drum was preheated before loading, which proved extremely beneficial.

The drum was initially set at a low speed of 7 rpm, which was increased to 10 rpm when 2000 g of the coating suspension had been applied. As confirmed with the factorial design results, these values were optimal to prevent poor uniformity of the coating color and to drastically minimize tablet erosion and white spots.

Finally, the atomization pressure was set at a low value of 2.5 bar, which eliminated issues with the coating color uniformity and reduced white spots, as established by Just, S. et al. [14], but which was the opposite to that found by Tobiska, S. et al. [18].

The results of the validation (see Table 6) confirmed that the batches obtained are of adequate quality and without significant aesthetic problems. This shows that the strategy defined in this study is suitable for quality improvements of commercial products already available on the market.

Two of the quality problems studied, erosion and inadequate coating uniformity, were completely eliminated in the validation batches. Regarding the white spots, these were considerably reduced compared to the commercial batches evaluated in the preliminary study. In addition, thanks to improvements in the process, the defects found were not considered significant, as it was very difficult to detect them visually.

It had also been possible with the validation to demonstrate that the results obtained in the factorial design with a placebo were reproducible in batches containing the active ingredient, with the consequent cost savings with this type of study. For this reason, it was considered more appropriate to carry out the full factorial design with placebo batches maintaining the same commercial batch size as, in this way, the behavior was much more representative of the actual manufacturing of commercial batches, and the same equipment was used.

However, it should be noted that the validation batches were manufactured with the active ingredient.

### 3.4. Industrial Batches

Since the optimization of the formula, 42 batches were manufactured, in which the usual defect analysis has been carried out. The results found appear in Table 7.

As the data show, the uniformity and erosion defects disappeared from the batches; however, the white spots defects appear sporadically, having more incidence in some batches (this fact has already been observed in the validation batches, see Table 6).

However, it should be noted that all batches have met the laboratory’s quality specifications and have not required additional controls or revisions, before being released to the market, regarding aesthetic defects.

## 4. Conclusions

Critical process parameters of the coating phase of a commercial product could be identified with a retrospective analysis of commercial batches. This demonstrates that existing processes can be improved by evaluating the historical data, thus ruling out the need to repeat product development tests.

The identification of process parameters, for the improvement of quality defects, may be established from a retrospective statistical analysis, ruling out the need for an experimental design with a battery of associated tests. The parameters established with this study are: 100% theoretical quantity of suspension applied, cooling time of 15 min, pre-heating time of 4 min, optimum spray pressure of 2.5 bar, and spray rate of 45 g/min.

Improvement techniques are established for the current process, from a technological point of view, which allowed the coating process to be optimized without making any changes to the manufacturing process and reducing aesthetic-type quality defects. Through the retrospectively evaluated data, a design of experiments could be conducted that allowed the minimum number of factors influencing process optimization to be selected.

Thanks to the factorial design and subsequent validation, a design space is established that allows a coating process to be obtained that guarantees a producible, safe, effective, and high-quality medicine. With the DoE, it has been possible to determine the optimal values for the product temperature and drum speed, which could not be established with the retrospective experimental design, to minimize or eliminate the associated quality defects. The models generated during the data analysis have been used to optimize the coating process with the goal of consistently meeting the defect release specifications. The parameters established with this study are: Optimum product temperature 44 °C and initial drum speed 7 rpm, increasing to 10 rpm when 2000 g of the coating suspension has been applied. The industrial validation demonstrated that these are the optimal parameters to obtain a product of adequate quality.

Therefore, direct and indirect costs of the tablet-coating process could be cut by improving the quality attributes of the medicine (safer, more efficient, and more stable) as a design space can be established with minimal investment in experiments, because retrospective data from commercial batches can be evaluated and analyzed statistically. The setup recommended by the software has given consistent results that passed specifications with little variability. The decision has been made to use this for production. It has provided excellent results ever since.

Although the duration of the coating process may be longer, indirect process costs are cut as there are no quality problems or subsequent corrective action.

Through commercial validation, the results clearly demonstrate that retrospective data analysis is a very useful tool in optimizing the coating processes of commercial products. From the optimization study, no more batches have been reprocessed or rejected with respect to the aesthetic defects, although the defect of white spots continues to be appreciated.

In conclusion, a research system has been developed that can be applied to the improvement of drug production processes. This type of study can, therefore, be used as a tool to optimize the robustness of existing processes, ensuring compliance with ICH Q8, Q9, and Q10 regulations, and also to identify critical parameters to be included and analyzed in the annual product quality review (PQR) of commercialized products.

## Figures and Tables

**Figure 1 pharmaceutics-12-00743-f001:**
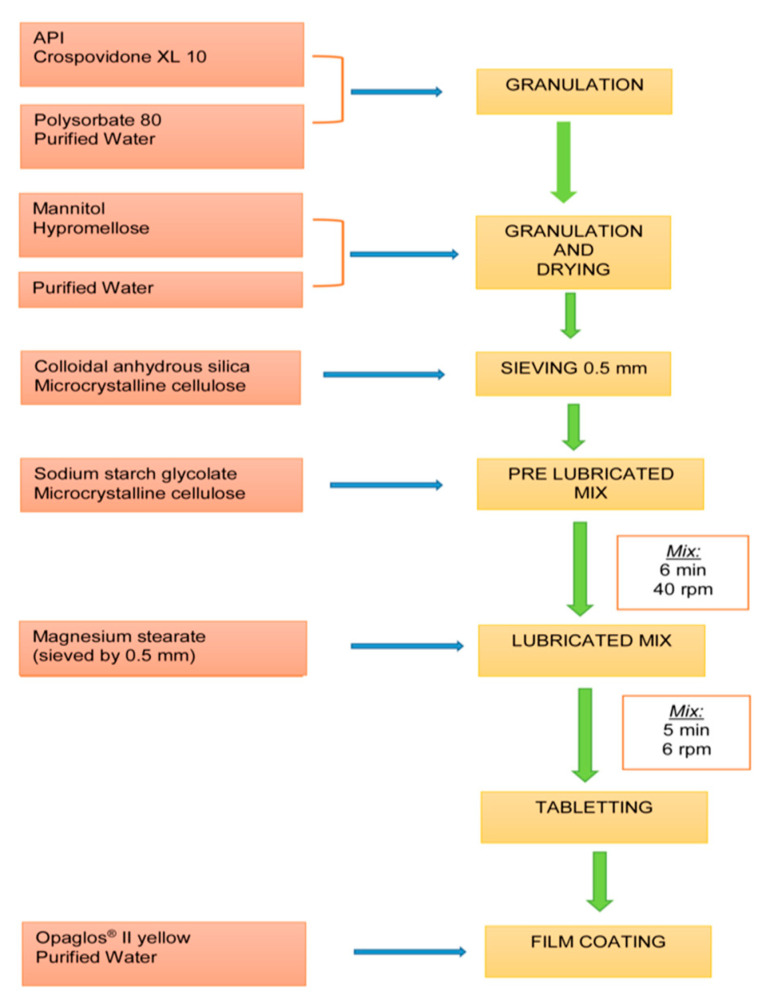
Industrial batch manufacturing process.

**Figure 2 pharmaceutics-12-00743-f002:**
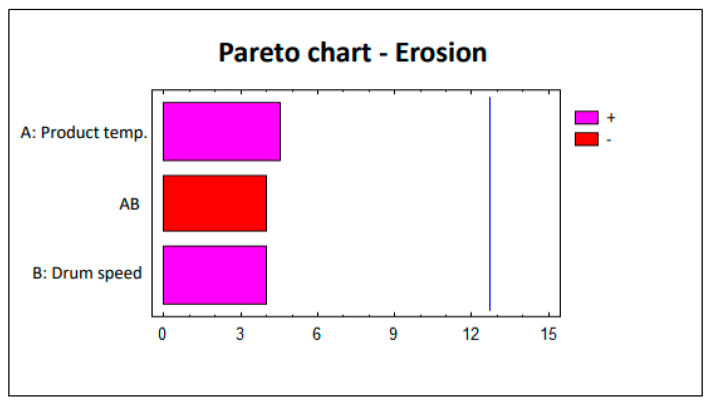
Study 1 Pareto chart—Erosion.

**Figure 3 pharmaceutics-12-00743-f003:**
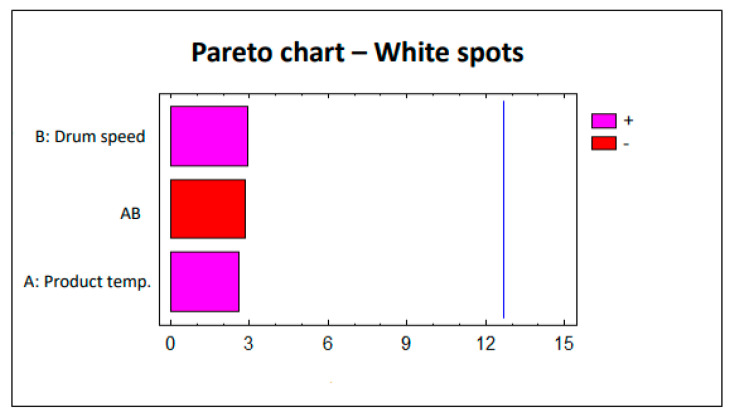
Study 1 Pareto chart—White spots.

**Figure 4 pharmaceutics-12-00743-f004:**
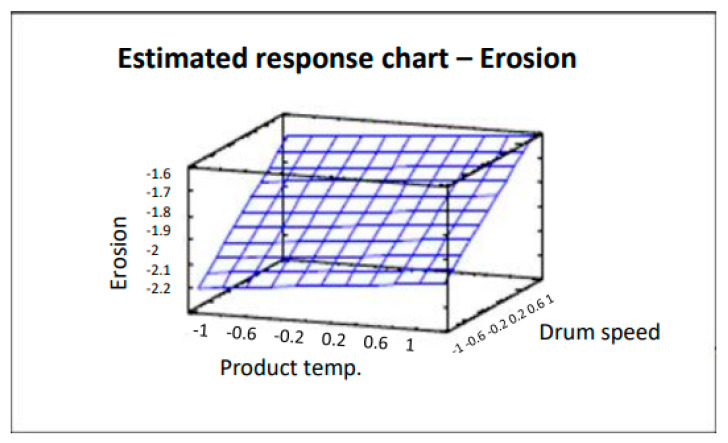
Estimated response chart—Erosion.

**Figure 5 pharmaceutics-12-00743-f005:**
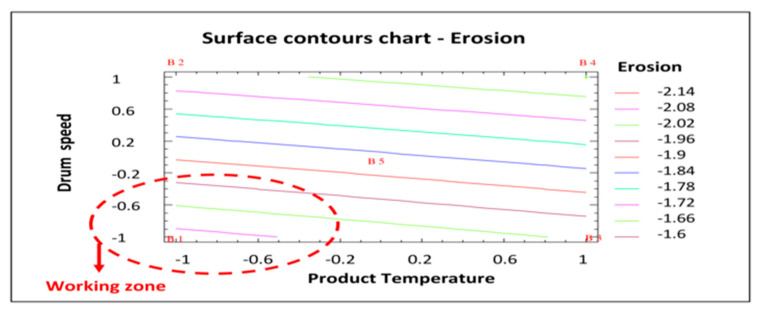
Surface contours chart—Erosion. B1: Show the conditions assayed of Batch 1, etc.

**Figure 6 pharmaceutics-12-00743-f006:**
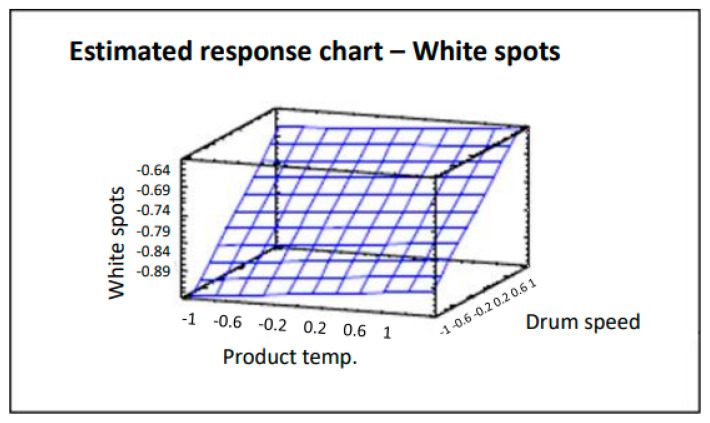
Estimated response chart—White spots.

**Figure 7 pharmaceutics-12-00743-f007:**
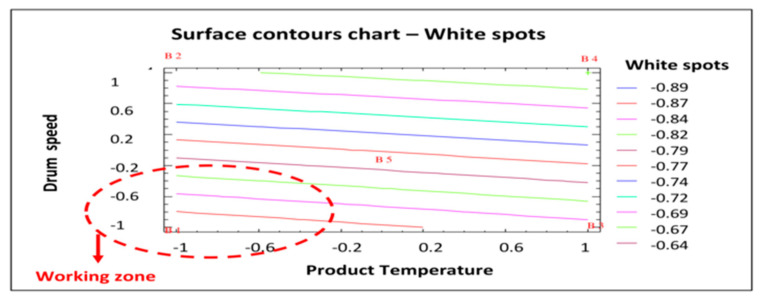
Surface contours chart—White spots. B1: Show the conditions assayed of Batch 1, etc.

**Table 1 pharmaceutics-12-00743-t001:** Proposed process parameter values according to the previous retrospective analysis of the process.

Variable/Parameter	New Ranges		
	Target	Min	Max
**Amount of dispersion (%)**	100	98	105
**Coating time (min)**	>150	150	According to process
**Cooling time (min)**	15	15	15
**Warm-up time (min)**	4	2	5
**Inlet temperature (°C)**	60	50	70
**Atomization pressure (bar)**	2.5	2.5	3
**Spray rate (g/min)**	45	40	50
**Product temperature (°C)**	To be determined ^+^	43	48
**Drum speed (rpm)**	To be determined ^+^	5	10

^+^ Conclusion from preliminary multivariate study and retrospective statistical analysis 4 was not clear for these two parameters, so a later 2^2^ prospective factorial design was planned.

**Table 2 pharmaceutics-12-00743-t002:** Prospective factorial design with the cytostatic drug (two factors to study).

Factorial Design 2^2^	Studied Variables			
	Product Temp.	Drum Speed	Product Temp. (°C)	Drum Speed (rpm)
**Trial 1**	−	−	44	7/10/10
**Trial 2**	−	+	44	9/12/14
**Trial 3**	+	−	48	7/10/10
**Trial 4**	+	+	48	9/12/14
**Central Point**			46	8/11/12

**Table 3 pharmaceutics-12-00743-t003:** Average values per process parameter in each manufactured batch. The process parameters to be optimized are marked in the table.

Variable	Batch 1	Batch 2	Batch 3	Batch 4	Central Batch
**Product temp. (°C)**	44	44	48	48	46
**Product temp. Min (°C)**	42	42	47	47	45
**Product temp. Max (°C)**	45	46	50	51	50
**Inlet temp. (°C)**	57	57	60	60	58
**Inlet temp. Min (°C)**	54	54	58	59	54
**Inlet temp. Max (°C)**	66	62	61	65	59
**Exhaust temp. (°C)**	45	45	49	49	47
**Exhaust temp. Min (°C)**	44	44	46	47	46
**Exhaust temp. Max (°C)**	46	48	50	51	49
**Atomization pressure (bar)**	2.5	2.5	2.5	2.5	2.5
**Drum speed 1 (rpm)**	7	9	7	9	8
**Drum speed 2 (rpm)**	10	12	10	12	11
**Drum speed 3 (rpm)**	10	14	10	14	12
**Airflow (m^3^/h)**	600	600	600	600	600
**Spray rate (g/min)**	45	45	45	45	45
**Warm-up time (min)**	5	5	5	5	5
**Coating time (min)**	213	220	260	256	275
**Drying (min)**	5	5	5	5	5
**Cooling (min)**	15	15	15	15	15
**Amount suspension (%)**	100.4	100.3	102.5	100.0	100.0
**Process stoppage (n°)**	1	1	1	2	5

**Table 4 pharmaceutics-12-00743-t004:** Visual inspection results (acceptance quality limit (AQL)) per batch.

Batch		Process Parameters		AQL 500 Tablets (n° Defective Tablets)		
		Product Temp. (°C)	Drum Speed (rpm)	Erosion	White Spots	Coating Uniformity
**Batch 1**	AGS.141201.01	44	7/10	0	1	0
**Batch 2**	AGS.141201.02	44	9/12/14	7	8	0
**Batch 3**	AGS.141217.01	48	7/10	13	2	0
**Batch 4**	AGS.141218.01	48	9/12/14	8	0	0
**CENTRAL**	AGS.141218.02	46	8/11/12	11	1	0

**Table 5 pharmaceutics-12-00743-t005:** Proposed process parameters for the validation batches.

Variable/Parameter	New Ranges		
	Target	Minimum	Maximum
**Product temperature**	44 °C	43 °C	46 °C
**Drum speed 1**	7 rpm	7 rpm	7 rpm
**Drum speed 2**	10 rpm	10 rpm	10 rpm
**Drum speed 3**	10 rpm	10 rpm	10 rpm

**Table 6 pharmaceutics-12-00743-t006:** Visual inspection results (AQL) for each validation batch.

Batch	AQL 500 Tablets (n° Defective Tablets)		
	Erosion	White Spots	Coating Uniformity
**Batch 1**	0	1	0
**Batch 2**	7	8	0
**Batch 3**	13	2	0
**Batch 4**	8	0	0

**Table 7 pharmaceutics-12-00743-t007:** Results of the quality control of the post-optimization batches elaborated and released to the market.

Commercial Batch	AQL 500 Tablets (n° Defective Tablets)		
	White Spots	Erosion	Coating Uniformity
C-1	1	0	0
C-2	0	0	0
C-3	2	0	0
C-4	1	0	0
C-5	5	0	0
C-6	3	0	0
C-7	20	0	0
C-8	4	0	0
C-9	11	0	0
C-10	24	0	0
C-11	17	0	0
C-12	6	0	0
C-13	1	0	0
C-14	1	0	0
C-15	6	0	0
C-16	6	0	0
C-17	0	0	0
C-18	0	0	0
C-19	2	0	0
C-20	3	0	0
C-21	4	0	0
C-22	1	0	0
C-23	1	0	0
C-24	0	0	0
C-25	8	0	0
C-26	0	0	0
C-27	3	0	0
C-28	3	0	0
C-29	4	0	0
C-30	0	0	0
C-31	5	0	0
C-32	4	0	0
C-33	2	0	0
C-34	0	0	0
C-35	0	0	0
C-36	12	0	0
C-37	8	0	0
C-38	1	0	0
C-39	0	0	0
C-40	1	0	0
C-41	2	0	0
C-42	15	0	0

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
