# Peer review of "Robustness Optimization of an Existing Tablet Coating Process Applying Retrospective Knowledge (rQbD) and Validation"

_pharmaceutics, 2020, doi:10.3390/pharmaceutics12080743_

Round 1
Reviewer 1 Report
The manuscript describes an interesting optimisation method for tablet coating by applying design of experiments to retrospective data combined with validation. Although the work is of interest, some aspects (especially with respect to the methods) are confusing and need more information and clarification. The following minor issues therefore need to be addressed:
1) Abtsract, line 24: second 2 should be superscript.
2) Page 2, line 50: Reference is made to 3 surface defects here, while only 2 (erosion and white spots) are described in line 86 on the same page. Please clarify by listing the three surface defects.
3) Page 2, line 53: Reference is made to historical data from four years, while on page 3, line 97 it is mentioned that the product is produced since 2007 - thus fourteen years?
4) Page 2, line 82: Reference is made to 36 batches as compared to 42 batches on page 11, line 360.
5) Page 4, line 116: Please clarify what is meant by "these batches". Please give more information on how many placebo batches were produced and a more detailed description on how they were used in the DoE? Was the active ingredient replaced with excipient?
6) Page 6, line 182: Please give the length of the interruption breaks.
7) Page 7, line: Please gibe more information on the "5 trials".
8) Page 8, line 221: Please change "lower" to "relatively low".
9) General recommendation: The discussions on the results should be changed to past tense (e.g. page 11, line 318 "produces" should be "produced" but there are many similar changes necessary).
Author Response
The manuscript describes an interesting optimisation method for tablet coating by applying design of experiments to retrospective data combined with validation. Although the work is of interest, some aspects (especially with respect to the methods) are confusing and need more information and clarification. The following minor issues therefore need to be addressed:
1) Abstract, line 24: second 2 should be superscript.
It was a typographic error and it has been modified in the main paper.
2) Page 2, line 50: Reference is made to 3 surface defects here, while only 2 (erosion and white spots) are described in line 86 on the same page. Please clarify by listing the three surface defects.
As it has been proposed by the reviewer, some information and some clarifications have been added to the main paper (line 50).
“Such surface defects can be classified into the following three groups: erosion, white spots and poor coating uniformity”.
3) Page 2, line 53: Reference is made to historical data from four years, while on page 3, line 97 it is mentioned that the product is produced since 2007 - thus fourteen years?
The product has been manufactured in this facility since 2007, but for the retrospective analysis the 4-year data were taken, since they were not computerized and had to be collected manually. A full 4 years was considered to be a sufficient sample.
4) Page 2, line 82: Reference is made to 36 batches as compared to 42 batches on page 11, line 360.
The first retrospective study (reference 4) was made based on 36 batches, from this study the parameters to be optimized and their intervals were concluded.
- Galí A, García-Montoya E, Ascaso Met al. Improving tablet coating robustness by selecting critical process parameters from retrospective data. Pharm Dev Technol. 2016;21(6):688‐697.
These 42 batches, line 360, are the number of batches manufactured after optimization of the process and after validation, so no optimization is made on them. They are simply reported as an indication that the optimization has been successful. As it was not necessary to change the registration presented to the regulatory authorities, these new set points were applied in the following industrial lots.
In this second analysis, 42 batches were taken for the study.
5) Page 4, line 116: Please clarify what is meant by "these batches". Please give more information on how many placebo batches were produced and a more detailed description on how they were used in the DoE? Was the active ingredient replaced with excipient?
“These batches” refers to the factorial design batches (5 placebo batches in total): 4 batches of the 22 + 1 central batch.
The active ingredient was offset by the majority excipient of the formulation: Mannitol (Pearlitol 160C), which is the excipient commonly used to compensate for the correction of the amount of API added in each manufacture according to the potency.
6) Page 6, line 182: Please give the length of the interruption breaks.
As it has been proposed by the reviewer, the length of the interruption breaks has been added in the main paper( it is approximately 5 seconds).
7) Page 7, line: Please give more information on the "5 trials".
As it has been proposed by the reviewer, some information have been added to the main paper (line 189).
“A sample of 500 coated tablets is taken from each batch (5 commercial-scale batches, details of process in table 2) for visual inspection to determine the number of defective tablets per batch. The results are shown in Table 4.”
8) Page 8, line 221: Please change "lower" to "relatively low".
It has been modified in the main paper.
9) General recommendation: The discussions on the results should be changed to past tense (e.g. page 11, line 318 "produces" should be "produced" but there are many similar changes necessary).
It has been modified in the main paper.

Reviewer 2 Report
It's a solid work. I have just minor remark about surface plots Figure 4 and Figure6. First it seems that we have linear relationships hre - it is necessary to comment on this in the text. Second: please provide visible experimental points on these plots to make it easier for interpretation. Good luck!
Author Response
It's a solid work. I have just minor remark about 1) surface plots Figure 4 and Figure 6. First it seems that we have linear relationships here - it is necessary to comment on this in the text.
In the work you can find the complete statistical analysis and more graphs that relate the factors studied: http://hdl.handle.net/2445/96035 (reference 16)
However, sentences with the correlation coefficient in the text (lines 222 and 242) and the reference (16), has been added in case that readers want to expand the details of the work.
“Analysis of variance gives an R2 correlation of 97.3%, more data and graphs are accessible in the full paper (16).”
“Analysis of variance gives an R2 correlation of 94.0 %, more data and graphs are accessible in the full paper (16).”
- Galí Serra A. Optimización en la fabricación de medicamentos según ICH Q8, Q9 y Q10: aplicación a comprimidos recubiertos mediante diseño experimental de datos retrospectivos [Internet]. Dipòsit Digital de la Universitat de Barcelona. Tesis Doctorals. 2015. Available from: http://hdl.handle.net/2445/96035
As it has been proposed by the reviewer, figures have been modified and added to the main document.

Reviewer 3 Report
Its an interesting to improve manufacturing process. The manuscript is written well and results are discussed and concluded with clarity.
Author Response
Thank you very much.